# Text Boosts Generalization: A Plug-and-Play Captioner for Real-World Image Restoration

## Abstract

Generalization has long been a central challenge in real-world image restoration. While recent diffusion-based restoration methods, which leverage generative priors from text-to-image models, have made progress in recovering more realistic details, they still encounter "generative capability inactivation" when applied to out-of-distribution data. To address this, we propose using text as an auxiliary invariant representation to reactivate the generative capabilities of these models. We begin by identifying two key properties of text input in diffusion-based restoration: richness and relevance, and examine their respective influence on model performance. Building on these insights, we introduce Res-Captioner, a module that generates enhanced textual descriptions tailored to image content and degradation levels, effectively mitigating response failures. Additionally, we present RealIR, a new benchmark designed to capture diverse real-world scenarios. Extensive experiments demonstrate that Res-Captioner significantly boosts the generalization ability of diffusion-based restoration models, while remaining fully plug-and-play.

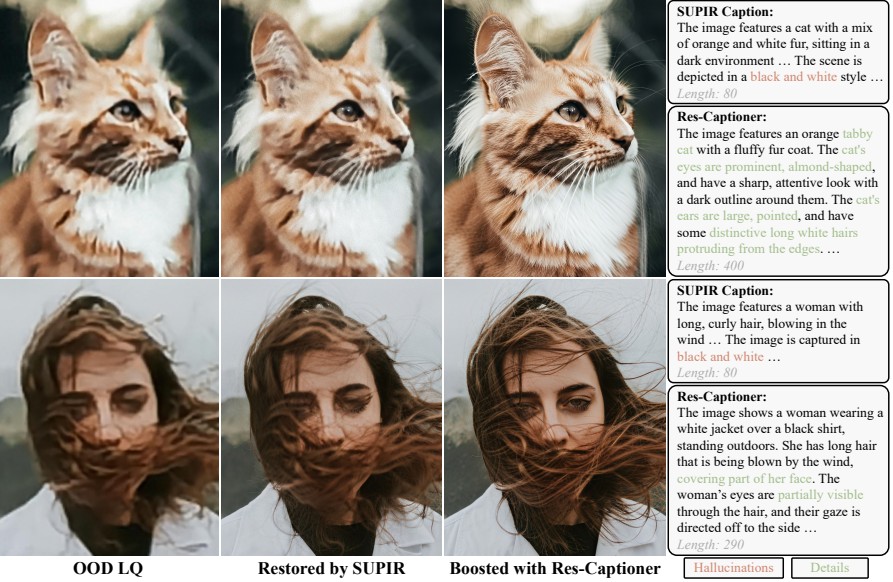

Figure 1: State-of-the-art methods like SUPIR (Yu et al., 2024a) face "generative capability inactivation" on out-of-distribution (OOD) data. Our Res-captioner reactivates their generative capabilities by providing detailed and accurate descriptions.

## 1 Introduction

Diffusion-based image restoration methods (Yu et al., 2024a; Sun et al., 2024; Wu et al., 2024; Wang et al., 2024b; Lin et al., 2023; Tao Yang & Zhang, 2023; Ai et al., 2024; Yu et al., 2024b; Zhang et al., 2024), powered by pre-trained text-to-image (T2I) models (Rombach et al., 2022; Podell et al., 2024), achieve superior texture and detail recovery compared to GAN-based methods (Zhang et al., 2021; Wang et al., 2021; Liang et al., 2021; 2022a; Chen et al., 2022). However, these models still

face the out-of-distribution (OOD) challenge (Koh et al., 2021), arising from misalignment between training data and real-world test cases. Real-world degradation simulations (Zhang et al., 2021; Wang et al., 2021) offer a common mitigation approach, but a domain gap persists (Liu et al., 2023b; Wang et al., 2024a; Kong et al., 2022), especially pronounced for device-induced degradations. As depicted in Figure 1, even state-of-the-art methods struggle to restore fine textures under complex degradations, a limitation we refer to as "generative capability deactivation".

We define image restoration as $x = \mathcal{R}(x_{lq})$, where $x$ and $x_{lq}$ denote high-quality (HQ) and low-quality (LQ) images, respectively, and $\mathcal{R}$ is the restoration model. To tackle domain generalization, researchers propose learning a cross-domain invariant representation $z = \mathcal{G}(x_{lq})$ (Arjovsky et al., 2019; Nguyen et al., 2021; Li et al., 2022a) and then train a prediction network conditioned on $z$: $x = \mathcal{H}(z)$. However, learning degradation-invariant representations with strong generalization and minimal information loss remains difficult in image restoration (Liu et al., 2022), as decoupling content from degradation in the image modality is challenging (Chen et al., 2024; Tran et al., 2021; Li et al., 2022b). To address this, we propose transforming LQ images into the text modality using an image captioner $\mathcal{C}$: $y = \mathcal{C}(x_{lq})$, leveraging recent multi-modal advancements (Liu et al., 2024b;a; Chen et al., 2023). This approach offers two advantages: first, in the text modality, degradation-related descriptions $y_{deg}$ can be easily separated, leaving the content-related part $y_{cont} = \{w \mid w \in y, w \notin y_{deg}\}$ as a degradation-invariant representation of $x_{lq}$. Second, text naturally activates priors in T2I diffusion models, facilitating enhanced texture recovery (Yu et al., 2024a; Sun et al., 2024; Wu et al., 2024; Tao Yang & Zhang, 2023; Yu et al., 2024b; Zhang et al., 2024).

However, due to significant information compression during the image-to-text transformation, relying solely on $y_{cont}$ cannot fully meet the high-fidelity requirements of image restoration tasks. Therefore, we utilize $y_{cont}$ as an auxiliary invariant representation in conjunction with the LQ image input, expressed as: $x = \mathcal{R}(x_{lq}, y_{cont})$. In our framework, image restoration is treated as a dual-conditioned image generation problem. Compared to the text input $y_{cont}$, the LQ image $x_{lq}$ serves as a much stronger condition, being more closely aligned with the final output $x$. However, when the degradation domain of the LQ image shifts, the information that the model can extract from $x_{lq}$ largely decreases, leading to the problem of generative capability deactivation (illustrated in Figure 1). To address OOD data, we propose adaptively enhancing the auxiliary invariant representation $y_{cont}$ through our Restoration Captioner (Res-Captioner), compensating for the information loss from $x_{lq}$ due to domain shifts.

To this end, we identify two key properties of text input in T2I diffusion-based restoration models: richness and relevance. Richness is primarily reflected in the length of the text; the more detailed the text, the richer the generated textures. Relevance, on the other hand, measures the correlation between the description and the HQ image content, with higher relevance leading to greater fidelity between the restored image and the ground truth. Building on these properties, we develop Res-Captioner, which is designed to accommodate varying degradation types and image clarity levels. Notably, Res-Captioner can be seamlessly integrated into restoration models, enhancing generalization without requiring retraining of the restoration model itself.

Finally, given the limitations of current real-world image restoration benchmarks (Cai et al., 2019; Wei et al., 2020), such as the restricted variety of imaging devices, and narrow content diversity, we introduce a new benchmark called **RealIR**. RealIR encompasses a broader range of degradation sources, clarity levels, and diverse photographic scenarios. Through this benchmark, we demonstrate that our Res-Captioner significantly improves the generalizability of diffusion-based methods, delivering more detailed and high-fidelity restoration results.

The contributions of this paper can be summarized as:

- We identify the potential of utilizing text as an ancillary invariant representation to enhance generalizability in image restoration, highlighting two key properties—richness and relevance—and their respective impacts on restoration performance.
- Building on our findings, we develop the Res-Captioner, which generates adaptively enhanced ancillary invariant representations, improving the generalizability of pre-trained diffusion-based restoration models in a plug-and-play fashion.
- We introduce a new restoration benchmark, RealIR, to comprehensively assess generalizability. Using both our benchmark and existing public datasets, we demonstrate the effectiveness of the Res-Captioner across multiple restoration methods.

## 2 RESTORATION CAPTIONER

### 2.1 PROPERTIES OF TEXT INPUT

We start by investigating how the text input $y$ affects the performance of restoration methods built on text-to-image (T2I) models. We identify two key properties of the text: richness and relevance. Richness refers to the amount of information conveyed, often reflected in text length, while relevance measures the degree of correlation between the text and the corresponding high-quality (HQ) image. Additionally, we observe that degradation-related or photography-specific descriptions can negatively affect restoration results, highlighting the importance of extracting content-specific descriptions, denoted as $y_{cont}$.

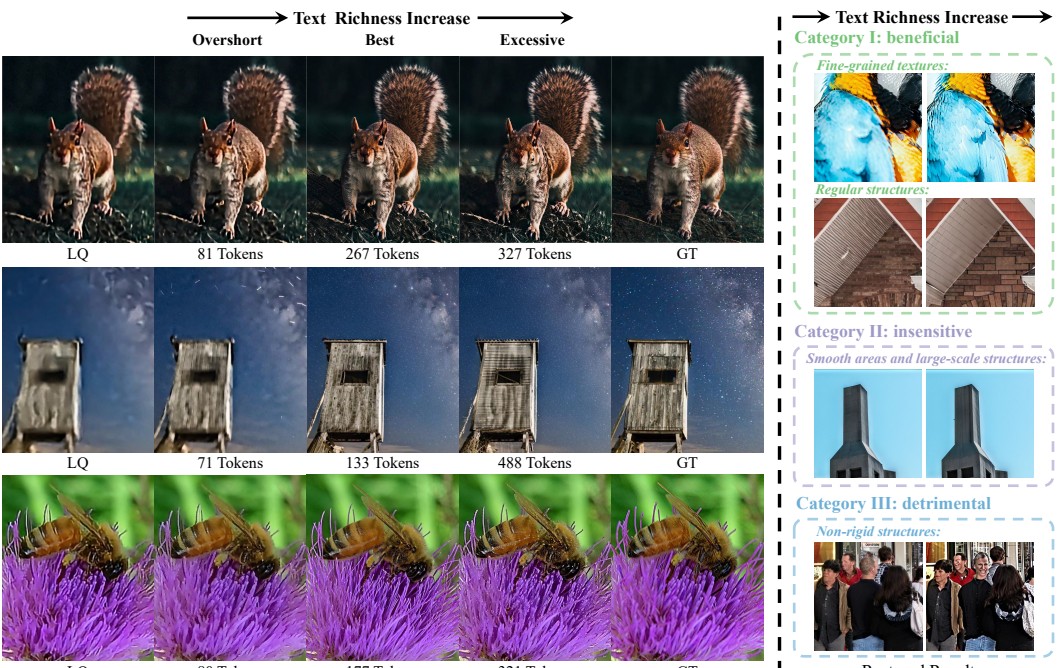

Figure 2: Visualization of the text richness property. (**Left**) The richness of textures and details in the restored results increases with text richness. Text that is too short can result in the "generative capability inactivation" problem. Excessively long text can lead to messy generation and artifacts. (**Right**) We can classify image content into three categories based on the effect of increased text richness: **I** beneficial, **II** insensitive, and **III** detrimental.

#### 2.1.1 RICHNESS PROPERTY

**Observation 1**. *The richness of restored textures and details increases proportionally with the richness of the text description.*

As illustrated in Figure 2, we observe that for all low-quality (LQ) images, increasing text richness (*i.e.*, text length) consistently enhances texture restoration. To explore this further, we prepare a dataset of 120 HQ images from diverse scenarios and generate the corresponding LQ images using Real-ESRGAN (Wang et al., 2021). GPT-4 is employed to generate detailed descriptions for the HQ images. We then evaluate two representative restoration models, SUPIR (Yu et al., 2024a) and StableSR (Wang et al., 2024b), for verification. The text input is encoded using CLIP (Radford et al., 2021) to generate 77 tokens. We then repeatedly append the last 20 tokens, excluding the EOS token, and follow (Xia et al., 2024) to integrate and inject these length-varying tokens into the restoration models to produce the restored results. Texture richness is assessed using two non-reference metrics: MANIQA (Yang et al., 2022) and MUSIQ (Ke et al., 2021). As shown in Figure 3 (**a, b**), both metrics demonstrate a positive correlation with the number of text tokens, supporting **Observation 1**.

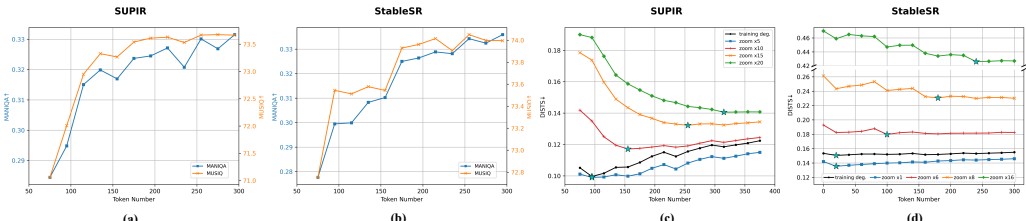

Figure 3: Demonstration of the richness property. **(a, b)**: There is a positive correlation between text richness and the richness of textures in the restored results. **(c, d)**: The optimal text richness (indicated by an asterisk) is proportional to the degree of deviation between the test degradation domain and the training degradation domain. Best viewed zoomed in.

We attribute this property to the data bias inherent in pre-trained T2I models, where images with richer content are typically paired with more detailed descriptions during training. Similar observations have been made in T2I research (Betker et al., 2023; Yang et al., 2024), where longer prompts lead to more enriched scenes. However, in the context of image restoration, this effect primarily enhances texture quality rather than introducing new objects or elements.

**Observation 2**. *The optimal level of text richness is influenced by factors such as degradation severity, and image content.*

As discussed, detailed text descriptions improve texture restoration. However, as shown in Figure 2, exceeding the optimal range of text richness may lead to undesirable artifacts or messy generation. For instance, the squirrel's eyes and mouth are misaligned with the LQ image, and the bee shows over-sharpening effects. We posit that the optimal text richness is proportional to the domain gap between training and testing degradations. To validate this, we prepare LQ images either simulated or captured in the wild with different zoom ratios and evaluate the performance of SUPIR and StableSR in relation to text richness. As illustrated in 3 **(c, d)**, as the test degradation increasingly diverges from the training setting (*e.g.*, $4\times$ Real-ESRGAN degradation), the optimal text richness similarly increases. This is because, as degradation severity intensifies, the useful information the model can extract from LQ images diminishes, necessitating more informative textual inputs to compensate for the information loss.

We also find that the optimal text richness is influenced by the content of the LQ image. Following (Liang et al., 2022b), we categorize three groups based on the impact of increased text richness on image content: beneficial, insensitive, and detrimental. Category I, "beneficial", includes fine-grained textures (*e.g.*, feathers, leaves, sand) and regular structures (*e.g.*, walls, windows), which benefit from longer text input as it activates the model's generative capability. Category II, "insensitive", consists of smooth areas and large-scale structures (*e.g.*, sky), where text richness has minimal effect. Category III, "detrimental", includes non-rigid structures (*e.g.*, text, crowds), where excessively long text may compromise fidelity.

### 2.1.2 RELEVANCE PROPERTY

**Observation 3**. *The fidelity of restored textures improves in correlation with the relevance of the text description.*

To quantitatively characterize the text relevance property, we introduce the concept of the "text-replacing ratio". This is defined as the ratio of original words in the text input $y$ that are replaced with non-meaningful words like "the" or "for". As the text-replacing ratio increases, the relevance between the text input and the corresponding HQ image decreases, while the text richness remains unchanged. As shown in Figure 4, we observe that although restored results retain rich textures as the text-replacing ratio increases, they suffer from a decline in fidelity. This is confirmed by the decreasing DISTS scores (lower is better), measured between the HQ image and the restored output. At higher text-replacing ratios, even the overall semantics of the restoration become distorted. For instance, at a ratio of 1.0, a lizard's head is incorrectly restored as flowers. In such cases, the model continues to generate textures but lacks the appropriate guidance to produce accurate ones.

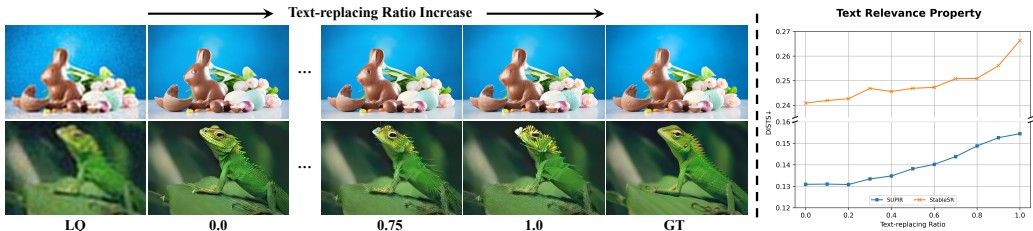

Figure 4: Visualization and demonstration of the text relevance property. **Left**: The accuracy of textures and details in the restored results decreases as the text-replacing ratio increases, indicating that text relevance contributes to the fidelity of the restoration. **Right**: DISTS increases with a higher text-replacing ratio, further indicating a decrease in the fidelity of the restored results.

### 2.1.3 HARMFUL DESCRIPTION

**Observation 4**. *Descriptions related to degradation or photography can lead to global or localized blurring in the restored images.*

We discover that degradation-related descriptions such as "blur" or "blurred", and photography terms like "shallow depth of field" or "bokeh effect", may lead to blurred outputs. Even when descriptions like "the background is blurred, while the main subject is sharp" accurately reflect the HQ image, they can cause overall blurring in the restored results. This is likely due to the limited spatial control capabilities of pre-trained T2I models (Avrahami et al., 2023), which amplifies the blurring effect. To validate this, we use GPT-4 to generate two captions of similar length: one without harmful descriptions and another including them. To exclude the effects of text richness and relevance, we duplicate the harmless description, labeled "Without Harmful Description", and combine both harmless and harmful descriptions to create "With Harmful Description". As shown in Figure 5, the description without harmful terms successfully restores clearer and richer details, while the harmful description does not.

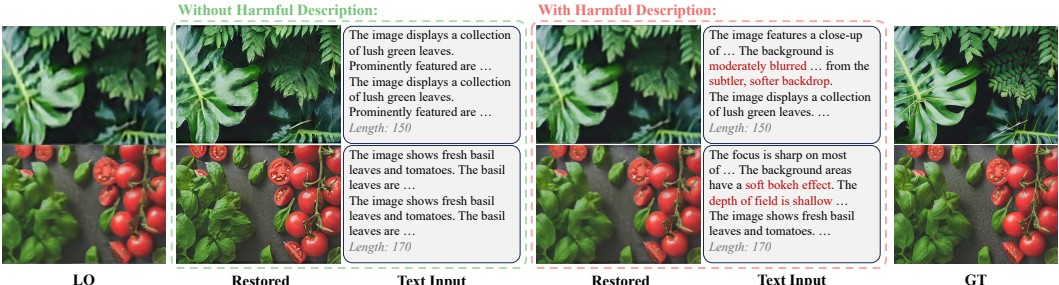

Figure 5: Harmful descriptions to the image restoration.

## 2.2 ANCILLARY INVARIANT REPRESENTATION ENHANCEMENT

Learning degradation-invariant representations from LQ images is highly challenging. To address this, we propose using text free of degradation-related descriptions as an auxiliary invariant representation to improve generalization. As discussed in Section 2.1, text plays a crucial role in controlling both the richness and fidelity of textures in restored results. However, existing image captioners (Liu et al., 2024b;a; Chen et al., 2023), which are not specifically designed for image restoration, not only generate harmful descriptions but also fail to adaptively enhance text richness. Consequently, they may contribute to the "generative capability inactivation" problem (Figure 1) in real-world scenarios. To address this issue, we introduce Res-Captioner, a restoration-specific captioner that generates high-quality text descriptions for real-world LQ images across diverse degradation levels and content categories, ensuring adaptive control over both richness and relevance.

**Training data generation.** We first collect HQ images from (Unsplash), ImageNet (Deng et al., 2009), and SAM (Kirillov et al., 2023), filtering out overly smooth ones using Sobel filters based on image gradient standard deviation. This ensures a selection of rich-content, high-clarity HQ images

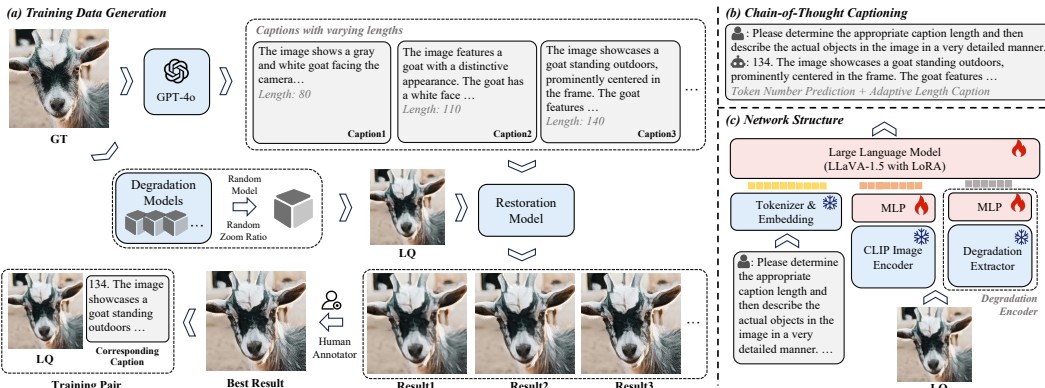

Figure 6: **(a)** The generation and annotation process of our training data. **(b)** Chain-of-Thought captioning of our Res-Captioner. **(c)** Network structure of our Res-Captioner.

from diverse scenarios. Next, as shown in Figure 6 (a), we leverage five pre-trained latent diffusion models (LDM) (Rombach et al., 2022) to generate LQ images that simulate varying imaging devices and zoom ratios. Training details for the LDM are provided in the appendix. We also include a percentage of Real-ESRGAN-generated (Wang et al., 2021) LQ images to further enhance diversity.

To ensure high relevance while minimizing hallucination, as illustrated in Figure 6 (a), we use GPT-4 to generate descriptions of varying lengths for each HQ image. Several prompting techniques, detailed in the appendix, are applied to avoid degradation-related or photography-specific content. These descriptions are fed into the restoration model, producing multiple restored candidates for each LQ image. Human annotators select the optimal text input that provides the best visual result, balancing texture richness and fidelity. The token length of the selected description is then calculated and combined with the description to form the final caption output in the format <token length, description>. In total, we curate 5,500 LQ image-caption pairs for training our Res-Captioner.

**Chain-of-Thought captioning.** Our goal is to generate accurate descriptions with appropriate richness for LQ images. As discussed in **Observation 1** and **Observation 2**, the optimal text richness, primarily reflected in token length, is determined by image content and degradation levels. Given the diversity of real-world scenarios, we enhance the reasoning and decision-making capabilities of Multimodal Large Language Models (MLLM) by adopting the "Chain of Thought" (CoT) strategy (Wei et al., 2022) in Res-Captioner. Specifically, as shown in Figure 6 (b), the model first predicts the optimal token number before generating the corresponding caption. As demonstrated in Section 3.4, this approach significantly improves the accuracy of the description length.

**Network structure.** We fine-tune LLaVA-1.5 (Liu et al., 2024a) using low-rank adaptation (LoRA) (Hu et al., 2021) to serve as our Res-Captioner. Since LLaVA is not designed for LQ images, we enhance its ability to perceive image degradations. In addition to the original CLIP visual encoder, we incorporate a degradation-aware visual encoder, as shown in Figure 6 (c). This encoder consists of a pre-trained degradation extractor, known for its sensitivity to various degradations (Chen et al., 2024; Liu et al., 2023a), and a lightweight adapter for improved degradation extraction. Specifically, the adapter is built from several MLP layers, first compressing the token count to 1 and then expanding it to N tokens (we set $N = 36$), enabling the encoder to focus on the global degradation representation while ignoring spatially varying content.

## 2.3 REALIR BENCHMARK

The current real-world restoration benchmarks (Cai et al., 2019; Wei et al., 2020) are limited by a narrow range of degradation types, insufficient diversity in imaging devices, and constrained content scope. To overcome these limitations, we introduce RealIR, a new benchmark featuring 152 real LQ images from eight imaging devices, including two DSLRs and six mobile phones, capturing images with varying zoom ratios. We also incorporate 53 LQ images sourced from the internet to capture degradations introduced by network transmission, which differ from device-specific degradations. The dataset covers a wide range of content, including portraits, animals, plants, and architectural scenes, enabling comprehensive evaluations of image restoration methods' generalizability.

## 3 EXPERIMENTS

### 3.1 IMPLEMENTATION DETAILS

Our Res-Captioner is built on LLaVA-1.5[1]. We train the model with a batch size of 128 over 500 steps using an A800 GPU, employing the Adam optimizer (Kingma & Ba, 2014) with a learning rate of $2 \times 10^{-4}$. We integrate Res-Captioners into two diffusion-based restoration models: SUPIR (Yu et al., 2024a), built on SDXL (Podell et al., 2024), and StableSR (Wang et al., 2024b), using Stable Diffusion 2.1 (Rombach et al., 2022). Our models operate in a plug-and-play fashion, seamlessly integrating with restoration models based on the same text-to-image (T2I) backbone. As in (Xia et al., 2024), we iteratively process the long text through the CLIP (Radford et al., 2021) text encoder.

Details of our training data generation and labeling process are provided in Section 2.2. We collect 5,500 low-quality (LQ) image-caption training pairs for SUPIR. Recognizing that different T2I backbones exhibit distinct text richness characteristics, we collect an additional 500 pairs for StableSR for fine-tuning. To match the resolution requirements of the respective T2I backbones, we resize the short edge of high-quality (HQ) images to 1024 for SUPIR and 512 for StableSR. The parameters for LoRA follow the standard LLaVA settings.

### 3.2 EXPERIMENTAL SETTINGS

**Test datasets.** Our proposed RealIR dataset encompasses diverse content and degradations from real-world scenes, making it ideal for assessing the generalization ability of restoration models. However, due to the absence of ground-truth images in RealIR, we create an additional multi-degradation test set comprising 120 LQ-HQ pairs using pre-trained latent diffusion models (LDM). To ensure fair evaluation, the degradations used for LQ generation are distinct from those in our training set. We categorize the LQ-HQ pairs into three degradation levels based on zoom ratio: light (zoom ratio of 3 to 7), moderate (zoom ratio of 8 to 10), and heavy (zoom ratio of 15 to 20). These two test sets enable a comprehensive evaluation of the restored results' detail richness and fidelity across varying degradation levels. Furthermore, we evaluate our approach on established real-world benchmarks such as RealSR (Cai et al., 2019) and DRealSR (Wei et al., 2020), using randomly cropped patches for more comprehensive analysis.

**Compared methods.** Our experiments include state-of-the-art (SOTA) real-world image restoration methods, such as GAN-based approaches like Real-ESRGAN+ (Wang et al., 2021) and DASR (Liang et al., 2022a), as well as diffusion-based models including StableSR (Wang et al., 2024b), SeeSR (Wu et al., 2024), CoSeR (Sun et al., 2024), and SUPIR (Yu et al., 2024a). Additionally, we compare our Res-Captioner with leading image captioning models, such as LLaVA-1.5 and ShareCaptioner (Chen et al., 2023).

**Evaluation metrics.** For test sets without ground truth, such as RealIR, we use non-reference evaluation metrics aligned with human perception, including MUSIQ (Ke et al., 2021), MANIQA (Yang et al., 2022), LIQE (Zhang et al., 2023), and NIQE (Zhang et al., 2015). For datasets with ground truth, we adopt perceptual distance metrics like DISTS Ding et al. (2020) and LPIPS Zhang et al. (2018), alongside the LIQE metric, which leverages large vision-language models for robust evaluation. Pixel-level metrics such as PSNR and SSIM are no longer considered, as they exhibit weak correlation with human perception, as discussed in related works (Yu et al., 2024a; Sun et al., 2024).

### 3.3 COMPARISON WITH STATE OF THE ARTS

#### 3.3.1 QUANTITATIVE RESULTS

Our quantitative results are organized into three parts. First, we assess the generalization ability of existing restoration methods in real-world scenarios using the RealIR benchmark, showing that Res-Captioner consistently improves their performance. Second, we evaluate the multi-degradation test set and the existing benchmarks, confirming that Res-Captioner not only enhances detail generation but also preserves fidelity across various degradation levels. Lastly, we compare the performance of other image captioners to our Res-Captioner, highlighting its superior effectiveness.

---

[1]https://huggingface.co/liuhaotian/llava-v1.5-13b

Table 1: Quantitative comparisons on our RealIR benchmark. We highlight **best** values for each metric and the results of Res-Captioner-enhanced models .

| Methods | RealIR (Cameras) | | | | RealIR (Internet) | | | |
|---|---|---|---|---|---|---|---|---|
| | MUSIQ↑ | MANIQA↑ | LIQE↑ | NIQE↓ | MUSIQ↑ | MANIQA↑ | LIQE↑ | NIQE↓ |
| Real-ESRGAN+ | 58.54 | 0.1784 | 2.425 | 5.646 | 58.34 | 0.2048 | 2.157 | 5.646 |
| DASR | 53.82 | 0.1487 | 2.208 | 6.748 | 50.84 | 0.1397 | 1.594 | 6.748 |
| CoSeR | 56.91 | 0.1163 | 2.597 | 4.042 | 66.67 | 0.1842 | 3.822 | 4.042 |
| SeeSR | 70.19 | 0.2138 | 3.768 | 3.749 | 72.65 | 0.2694 | 4.243 | 3.749 |
| StableSR | 66.15 | 0.1924 | 3.466 | 4.033 | 67.66 | 0.2012 | 3.913 | 4.033 |
| StableSR w/ Ours | 68.04 | 0.1955 | 3.615 | 3.888 | 70.90 | 0.2251 | 4.252 | 3.888 |
| SUPIR | 60.43 | 0.1651 | 2.983 | 3.492 | 71.94 | 0.2727 | 4.425 | 3.492 |
| SUPIR w/ Ours | **71.38** | **0.2543** | **4.056** | **3.389** | **73.26** | **0.3055** | **4.578** | **3.389** |

Table 2: Quantitative comparisons between the official model and the Res-Captioner-enhanced model under different degradation levels. We show the improvement percentage on each metric.

| Methods | Light Degradation | | | Moderate Degradation | | | Heavy Degradation | | |
|---|---|---|---|---|---|---|---|---|---|
| | DISTS↓ | LPIPS↓ | LIQE↑ | DISTS↓ | LPIPS↓ | LIQE↑ | DISTS↓ | LPIPS↓ | LIQE↑ |
| StableSR | 0.1791 | 0.3311 | 3.699 | 0.1864 | 0.3209 | 3.603 | 0.2181 | 0.4008 | 3.047 |
| StableSR w/ Ours | **0.1661** | **0.3222** | **3.735** | **0.1692** | **0.3086** | **3.857** | **0.1918** | **0.3773** | **3.604** |
| | 7.3% | 2.7% | 1.0% | 9.2% | 3.8% | 7.1% | 12.1% | 5.9% | 18.3% |
| SUPIR | 0.1821 | 0.3444 | 3.148 | 0.1883 | 0.3473 | 3.349 | 0.2159 | 0.4106 | 2.840 |
| SUPIR w/ Ours | **0.1680** | **0.3178** | **4.011** | **0.1621** | **0.3052** | **4.226** | **0.1873** | **0.3754** | **3.991** |
| | 7.7% | 7.7% | 27.4% | 13.9% | 12.1% | 26.2% | 13.3% | 8.6% | 40.5% |

Table 3: Quantitative comparisons on RealSR and DRealSR datasets. **Bold**: Best results.

| Methods | RealSR | | | DRealSR | | |
|---|---|---|---|---|---|---|
| | DISTS↓ | LPIPS↓ | LIQE↑ | DISTS↓ | LPIPS↓ | LIQE↑ |
| SUPIR | 0.2660 | 0.3889 | 3.477 | 0.2906 | 0.4741 | 3.655 |
| SUPIR w/ Ours | **0.2474** | **0.3667** | **4.081** | **0.2699** | **0.4409** | **4.208** |

Table 4: Quantitative comparisons of image captioners on restoration.

| Methods | DISTS↓ | LPIPS↓ |
|---|---|---|
| Llava-1.5 | 0.1886 | 0.3600 |
| ShareCaptioner | 0.1780 | 0.3394 |
| Res-Captioner | **0.1725** | **0.3328** |

**RealIR benchmark.** The results, shown in Table 1, evaluate both real LQ images captured by various cameras and LQ images collected from the internet. Overall, diffusion-based models exhibit superior visual quality compared to GAN-based models, due to their stronger generative capabilities. Notably, when integrated with our Res-Captioner, diffusion models such as StableSR and SUPIR show significant improvements across all metrics. This highlights how our approach fully activates the generative power of T2I-based restoration models for diverse real-world LQ images.

However, the improvement introduced by Res-Captioner varies across different restoration models. For example, Res-Captioner enhances the LIQE score of StableSR on manually captured RealIR data by approximately **4.3%**, while it increases the LIQE score of SUPIR by an impressive **36%**. We attribute this discrepancy to the differing generative capabilities of T2I models. In particular, SUPIR, based on SDXL, suffers from the "generative capability deactivation" issue, which is effectively reactivated by our Res-Captioner, unlocking its full potential.

**Fidelity evaluation.** We compare the original models with their Res-Captioner-enhanced versions on the multi-degradation test set, as shown in Table 2. For both StableSR and SUPIR, Res-Captioner consistently improves fidelity, demonstrated by significant gains in reference-based metrics like DISTS and LPIPS. Notably, the performance improvements increase with the severity of degradation. For instance, the DISTS score of enhanced StableSR improves by approximately **7.3%**, **9.2%**, and **12.1%** for light, moderate, and heavy degradation, respectively. This trend supports our approach of using text as an auxiliary invariant representation. As test degradation diverges further from the training distribution, the restoration model extracts less useful information from the LQ image, making the supplementary text provided by Res-Captioner increasingly beneficial.

Given the relatively simple and light degradation in RealSR and DRealSR, we use SUPIR as the reference model for evaluation. Our Res-Captioner significantly improves the performance of SUPIR in Table 3, further demonstrating the robustness of our approach in real-world scenarios.

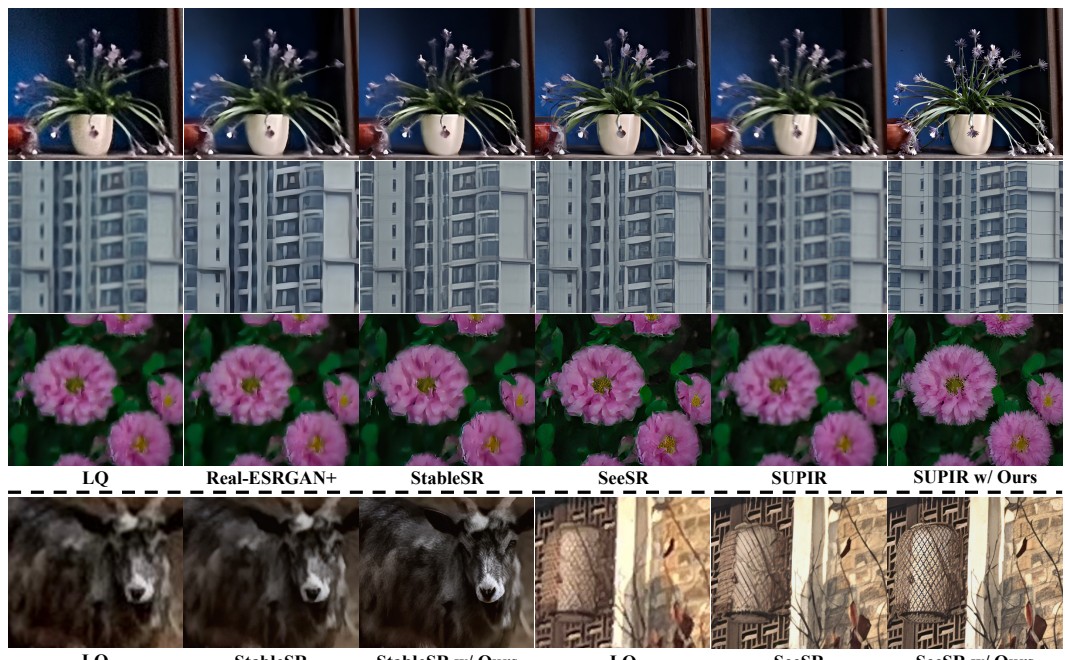

| LQ | Real-ESRGAN+ | StableSR | SeeSR | SUPIR | SUPIR w/ Ours |

| LQ | StableSR | StableSR w/ Ours | LQ | SeeSR | SeeSR w/ Ours |

Figure 7: Qualitative comparisons on in-the-wild images. **Upper**: Comparisons between SOTA restoration methods and Res-Captioner-enhanced SUPIR. **Lower**: Visual quality improvements introduced by Res-Captioner on StableSR and SeeSR.

**Comparison of image captioners.** We compare our Res-Captioner to LLaVA-1.5 and ShareCaptioner in a plug-and-play manner, integrating all captions into SUPIR as described in Section 3.1. Results from the multi-degradation test set, shown in Table 4, demonstrate that Res-Captioner provides superior guidance for image restoration. LLaVA-1.5 typically generates shorter captions (average length of 80), while ShareCaptioner consistently produces overly long captions (average length of 200). As noted in **Observation 2**, both overly short and excessively long captions can negatively affect restoration results. In contrast, Res-Captioner dynamically adjusts text richness based on the input image, optimizing restoration quality across varying degradation levels.

Beyond text richness, the generated descriptions differ significantly in quality. As discussed in the appendix, other captioners may produce misleading descriptions or hallucinations that degrade restoration quality, while our method generates highly relevant, accurate descriptions aligned with HQ images, effectively enhancing restoration results.

### 3.3.2 QUALITATIVE RESULTS

We provide visual comparisons on in-the-wild LQ images in Figure 7. In the upper section, Real-ESRGAN+ struggles with its limited generative capability, failing to recover high-definition textures. Both SUPIR and StableSR experience "generative capability deactivation" when handling out-of-distribution (OOD) data, leading to large areas of blurring. Although SeeSR responds better to OOD data, the textures it generates tend to appear overly smooth and unrealistic. In contrast, our Res-Captioner fully activates the generative potential of the T2I backbone in SUPIR, enabling the recovery of clearer, more realistic textures, such as detailed flower petals and building facades.

The lower section of Figure 7 illustrates how Res-Captioner improves other restoration models. When integrated with Res-Captioner, StableSR and SeeSR demonstrate an enhanced ability to recover fine-grained textures and structures, such as goat fur and lantern mesh, significantly outperforming their original versions. Notably, Res-Captioner can be directly applied to restoration models using the same T2I backbone without requiring fine-tuning.

Table 5: Ablation studies on text richness, relevance, and harmful descriptions. **Bold**: Best results.

| Method | Light Degradation | | Moderate Degradation | | Heavy Degradation | |
|---|---|---|---|---|---|---|
| | DISTS↓ | LPIPS↓ | DISTS↓ | LPIPS↓ | DISTS↓ | LPIPS↓ |
| Ours | **0.1680** | **0.3178** | **0.1621** | **0.3052** | 0.1873 | **0.3754** |
| w/ Min Len. | 0.1718 | 0.3274 | 0.1753 | 0.3252 | 0.2033 | 0.4009 |
| w/ Max Len. | 0.1864 | 0.3525 | 0.1770 | 0.3184 | 0.1964 | 0.4039 |
| w/ Low Rel. | 0.1738 | 0.3389 | 0.1655 | 0.3061 | 0.1907 | 0.3914 |
| w/ Harmful Des. | 0.1686 | 0.3191 | 0.1678 | 0.3178 | **0.1868** | 0.3883 |

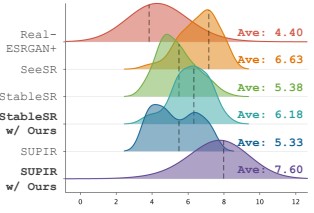

Figure 8: User study.

### 3.3.3 USER STUDY

To further validate Res-Captioner's ability to enhance generalization in real-world scenarios, we conduct a user study on in-the-wild LQ images with 31 experienced researchers. Each participant rates the visual perceptual quality (on a scale of 1 to 10, where higher is better) of results generated by Real-ESRGAN+, SeeSR, StableSR, StableSR with Res-Captioner, SUPIR, and SUPIR with Res-Captioner. As illustrated in Figure 8, StableSR and SUPIR show significant performance improvements when paired with Res-Captioner. Notably, SUPIR, when enhanced with Res-Captioner, delivers the highest visual quality among all methods.

### 3.4 ABLATION STUDY

We investigate the impact of the proposed text properties—richness, relevance, and harmful descriptions—on restoration performance by ablating each aspect in experiments. All models are trained under identical settings, with the only variation being the training data. Additionally, we analyze the effect of our proposed Chain-of-Thought (CoT) captioning and degradation-aware visual encoder on text richness. SUPIR is used as the restoration model in this section.

**Text richness.** To explore the impact of text richness, we create two training sets using the shortest and longest captions generated by GPT-4, corresponding to the results of "w/ Min Len." and "w/ Max Len." in Table 5. The results show that Res-Captioner achieves the best performance under varying degradation conditions due to its adaptive text richness capability. Moreover, we observe that the "w/ Max Len." model begins to outperform the "w/ Min Len." model as degradation severity increases, which is consistent with our **Observation 2**.

**Text relevance.** To study this property, we first calculate the length of human-selected optimal captions generated by GPT-4. We then produce same-length low-relevance captions using LLaVA-1.5 for training, denoted as "w/ Low Rel.". In contrast, our Res-Captioner ("Ours") achieves superior restoration results, highlighting the importance of high-relevance descriptions for restoration.

**Harmful descriptions.** In Section 2.1.3, we identify harmful descriptions that result in blurring in the restored images. Using the optimal text richness, we employ GPT-4 to generate captions incorporating these harmful descriptions. We fine-tune our Res-Captioner with this data, referred to as "w/ Harmful Des." in Table 5. The results show that harmful descriptions negatively affect restoration performance, causing an average **2.7%** decrease in LPIPS.

**CoT captioning and degradation-aware visual encoder.** We manually annotate the optimal text length, $L_o$, for 100 LQ images from the RealIR and multi-degradation datasets. To quantify the prediction error, we define the offset level $E$ as: $E = \max\left(|L_o - L| - 15, 0\right)/30$, where $L$ is the captioner's output length. The mean offset level for our Res-Captioner is 1.27. Without the CoT captioning, the mean offset level increases by **66.7%**, and without the degradation-aware visual encoder, it rises by **31.5%**. These results highlight the effectiveness of our model's design.

## 4 CONCLUSION

We leverage text as an auxiliary invariant representation to enhance the generalizability of T2I-diffusion-based restoration models. By focusing on two key properties of text inputs—richness and relevance—we propose Res-Captioner, which significantly improves real-world restoration performance in a plug-and-play manner.

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

# A APPENDIX

## A.1 REAL-WORLD LQ IMAGE GENERATION

By reproducing high-definition images, we collect numerous real-world LQ-HQ pairs for training the real-world LQ generation model. Data is gathered from five different devices, and five LQ generation models are trained to represent different types of degradation.

We select the latent diffusion model (LDM) (Rombach et al., 2022) as our LQ generator, training it to produce LQ images conditioned on corresponding HQ images. Additionally, the zoom ratio used during image reproduction is incorporated as another part of the conditional information.

For each degradation model, we retain one zoom ratio for the generation of multi-degradation test set, and the rest are used to generate Res-Captioner training data.

## A.2 DETAILS OF TRAINING DATA GENERATION

We use the following prompt to generate captions of varying lengths with GPT-4, while avoiding harmful descriptions through the use of restrictive phrasing.

```
Please describe the actual objects in the image in a very detailed
manner.  Please do not include descriptions related to the focus
and bokeh of this image.  Please do not include descriptions like
the background is blurred.  Please be careful to limit your answer
to about XXX words.
```

We generate a total of seven different caption lengths: 80 words, 110 words, 140 words, 200 words, 260 words, 350 words, and 440 words. The interval between lengths increases progressively, as longer captions tend to cause smaller texture changes when recovering with the same richness interval.

When training and testing the Res-Captioner, we use the following prompt:

```
Please determine the appropriate caption length and then describe
the actual objects in the image in a very detailed manner.  Please
do not include descriptions related to the focus and bokeh of this
image.  Please do not include descriptions like the background is
blurred.
```

## A.3 CONTENT QUALITY COMPARISON OF IMAGE CAPTIONS

In Figure 10 and 11, we compare the caption content quality between our proposed Res-Captioner and existing image captioners. Each caption was carefully examined, with hallucinations and harmful descriptions that could negatively impact restoration clearly marked. We also compared captions generated by Res-Captioner and other image captioners for the same image, highlighting the details missing in the others. Our method clearly demonstrates fewer hallucinations, is free from harmful descriptions, and produces a significant amount of detail closely aligned with the image content, which greatly supports the restoration process.

## A.4 MORE QUALITATIVE RESULTS

In this section, we provide additional visual comparisons between our method and state-of-the-art (SOTA) methods. As illustrated in Figure 12, when paired with the SUPIR restoration method, which features a powerful generative model backbone, our Res-Captioner shows clear advantages in recovery performance compared to previous SOTA methods. Additionally, our approach significantly improves visual quality when applied to the StableSR restoration method, as demonstrated in Figure 9, highlighting the robustness of our approach across different restoration models.

|      LQ      |    StableSR    | StableSR w/ Ours |      LQ      |    StableSR    | StableSR w/ Ours |

Figure 9: Additional qualitative comparisons of Res-Captioner applied to StableSR on in-the-wild images.

## A.5 ANALYSIS ON DEGRADATION TOKEN NUMBER

We analyze the impact of the token length $N$ in our degradation-aware visual encoder on text richness prediction accuracy. As shown in Table 6, the token length is inversely proportional to the text richness offset level $E$, indicating a positive relationship with text richness prediction accuracy. Balancing computational cost with performance benefits, we select $N = 36$ as the final degradation token number.

Table 6: Analysis of the token length in the degradation-aware visual encoder.

| Token Length $N$ | Offset Level $E$ |
| --- | --- |
| 4 | 1.53 |
| 9 | 1.46 |
| 16 | 1.34 |
| 25 | 1.28 |
| **36** | **1.27** |

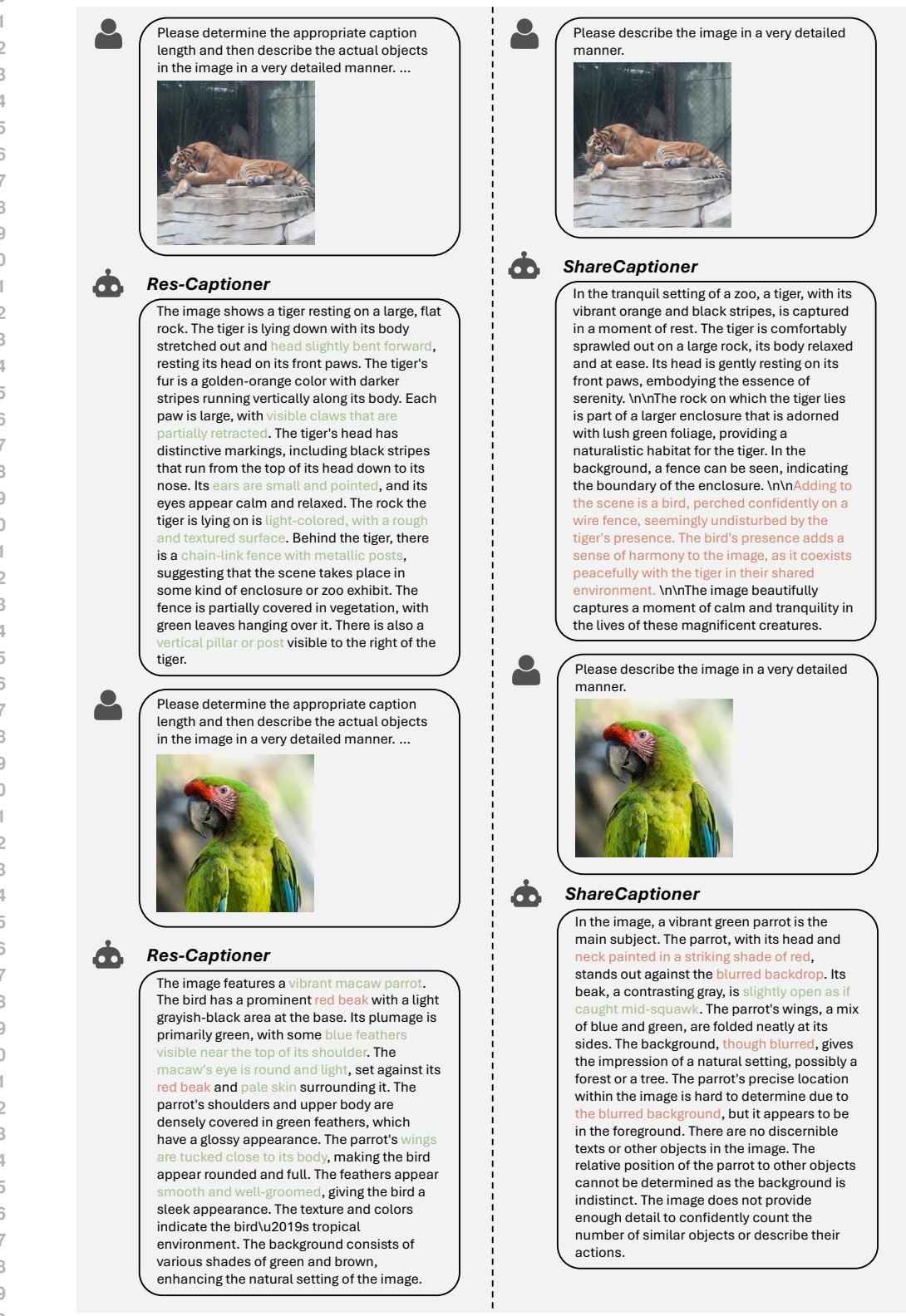

Figure 10: Content quality comparison between our proposed Res-Captioner and ShareCaptioner. We use red to indicate some hallucinations and harmful descriptions in the caption. We use green to highlight the detailed descriptions provided by one captioner that are missing in the other.

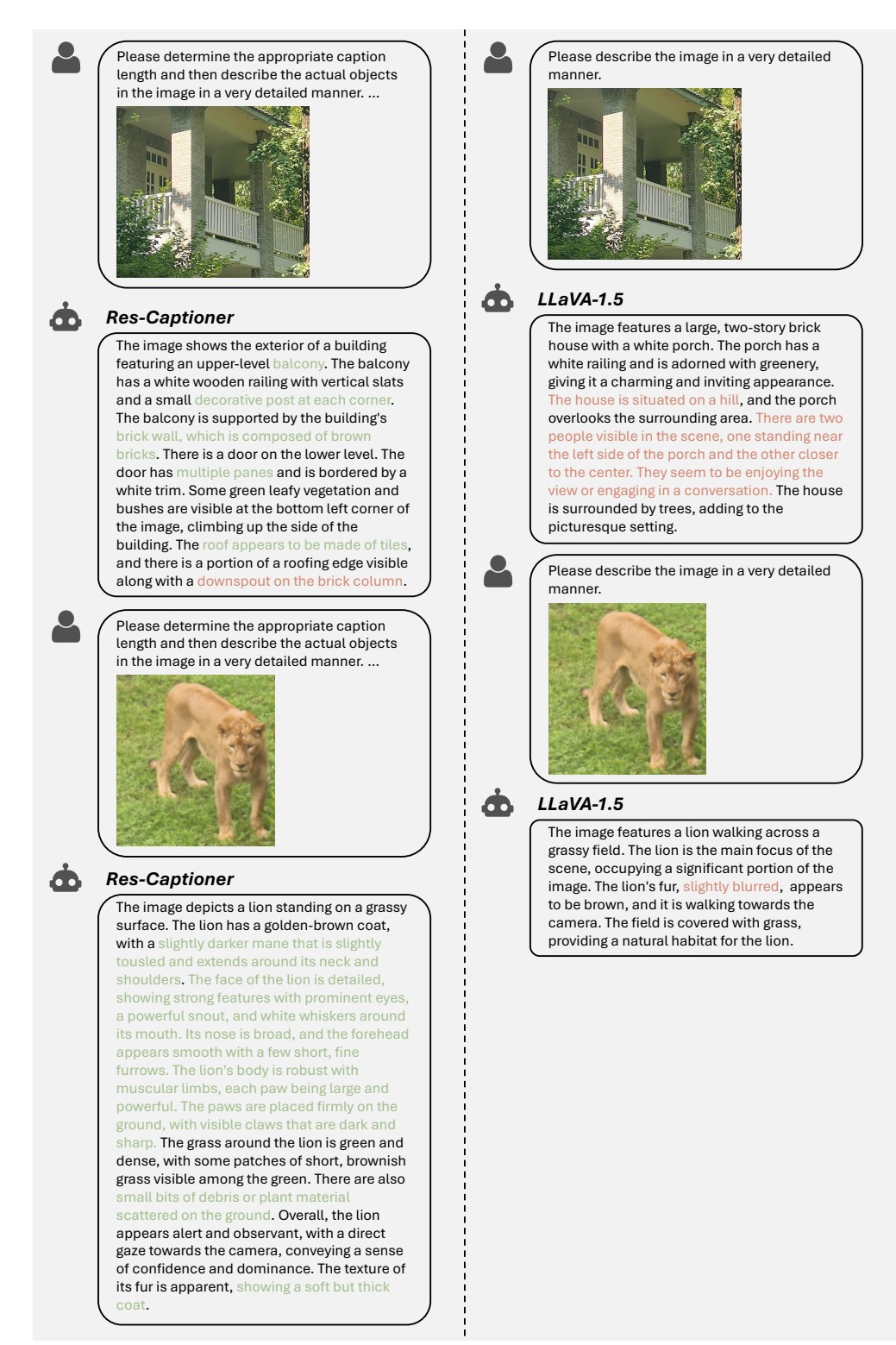

Figure 11: Content quality comparison between our proposed Res-Captioner and LLaVA-1.5. We use red to indicate some hallucinations and harmful descriptions in the caption. We use green to highlight the detailed descriptions provided by one captioner that are missing in the other.

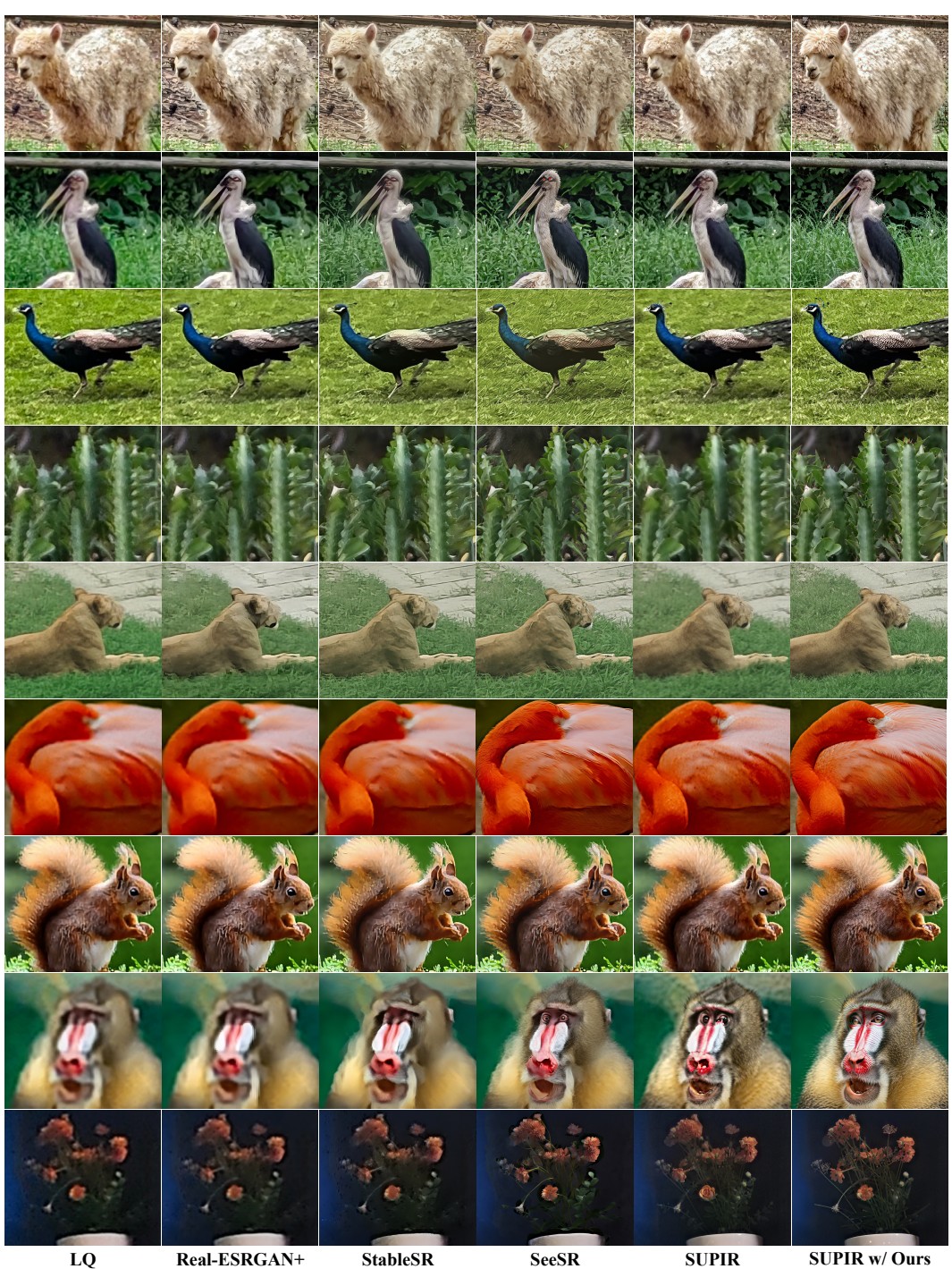

|      |                |          |        |        |              |
|------|----------------|----------|--------|--------|--------------|
| **LQ** | **Real-ESRGAN+** | **StableSR** | **SeeSR** | **SUPIR** | **SUPIR w/ Ours** |

Figure 12: Additional qualitative comparisons with SOTA methods on in-the-wild images.

