# OpenReview forum: "Text Boosts Generalization: A Plug-and-Play Captioner for Real-World Image Restoration"
_ICLR.cc/2025/Conference — ICLR 2025 Conference Withdrawn Submission_

### Official Review · Reviewer_dc9G · 2024-10-29

**Soundness:** 3
**Presentation:** 3
**Contribution:** 2
**Rating:** 5
**Confidence:** 5

**Summary:**

To improve the generalizability of pre-trained diffusion-based restoration models, the paper proposes a Res-Captioner to generate adaptively enhanced ancillary invariant text representations, and then injects the representations into diffusion models in a plug-and-play fashion.

**Strengths:**

1. The paper explores the impact of text richness and relevance on the generalization of the diffusion-based restoration model.
2. The paper constructs a Res-Captioner to generate adaptively enhanced ancillary invariant representations.
3. The paper introduces a new restoration benchmark to assess generalizability.

**Weaknesses:**

1. In text-to-image generation fields, some works[1,2,3] have noticed the importance of text encoders. The LQ image encoders in the image restoration fields correspond to the text encoders in text-to-image generation fields. The paper needs to add more discussion on them. In addition, I'm interested in the effect when applying ELLA[3] in a diffusion-based restoration model.
2. Recent image restoration work[4] also proposes to improve image restoration by removing degradation in the textual representations of a given degraded image. Some ideas in this paper are similar to it. The authors can add more discussion.
3. The paper can refer to the dataset and evaluation settings of SeeSR, giving more comparison metrics on more benchmarks and datasets. The paper also can utilize the datasets from [5] for evaluation. It can better demonstrate the effectiveness of the proposed method.
3. Most of the given image examples only have single objects and simple scenes. When the scenes of LQ images become more complex, are some of the observations and conclusions still applicable? You can try the datasets from [5] for evaluation.
4. Res-Captioner does not bring significant improvement compared to ShareCaptioner in Table 4.
5. The effect of the degradation extractor in Res-Captioner need to be experimented.


[1] Parrot: Pareto-optimal Multi-Reward Reinforcement Learning Framework for Text-to-Image Generation. https://arxiv.org/abs/2401.05675.
[2] Enhancing Diffusion Models with Text-Encoder Reinforcement Learning. https://arxiv.org/abs/2311.15657.
[3] ELLA: Equip Diffusion Models with LLM for Enhanced Semantic Alignment. https://arxiv.org/abs/2403.05135.
[4] Improving Image Restoration through Removing Degradations in Textual Representations. https://arxiv.org/abs/2312.17334.
[5] NTIRE 2024 Restore Any Image Model (RAIM) in the Wild Challenge. https://arxiv.org/abs/2405.09923

**Questions:**

Please see 'Weaknesses'. Limited discussion and experiments are my main concerns. I would increase the rating if the questions are addressed.

---

### Official Review · Reviewer_YHw2 · 2024-11-02

**Soundness:** 4
**Presentation:** 4
**Contribution:** 3
**Rating:** 6
**Confidence:** 2

**Summary:**

Res-Captioner offers improved text alignment, enhancing the richness and relevance of descriptions when reconstructing degraded images.
The method introduces a training strategy for a "degradation-robust description generator," which effectively strengthens real-world image restoration diffusion models.

**Strengths:**

1. Comprehensive studies on text conditioning in the restoration of degraded real-world images are novel and valuable.

2. The method demonstrates robustness to various degradations, making it applicable in a plug-and-play manner.

3. It effectively enhances the image restoration module, achieving state-of-the-art restoration performance.

**Weaknesses:**

1. Although text generation from degraded images is valuable, the restoration module or trained method remains limited in its applicability to various types of degradation. In most cases presented in this paper, the degraded images retain well-preserved low-frequency components. However, if certain regions are masked or affected by motion blur, the method may not adequately address the degradation.

2. The method generates training data using the restoration model to enhance the performance of that same model. If different restoration models are used for training data generation and evaluation, the performance of the restoration model may vary.

**Questions:**

1. SUPIR obtains text input by first providing an initially refined image (the input is processed through a degradation-robust encoder). In the case of application in SUPIR, can the method obtain text input without using this module?

2. What happens when the degraded image includes unseen conditions, such as inpainting or motion blur?

3. Which restoration model is used for training data generation, assuming a single model is applied across all experiments?

---

### Official Review · Reviewer_yBdk · 2024-11-02

**Soundness:** 2
**Presentation:** 3
**Contribution:** 2
**Rating:** 3
**Confidence:** 5

**Summary:**

This paper aims the problem of generalization in real-world image restoration. To address this, the authors suggest using richer and more relevant text prompt to boost generative capabilities of generative restoration models. They introduce Res-Captioner, a plug-and-play module that creates enhanced text descriptions tailored to image content and quality, helping models respond better. They also introduce RealIR, a benchmark for diverse real-world cases.

**Strengths:**

1. This article has made many attempts at the effects of different prompts on generative restoration models, and has summarized several principles. This is beneficial for the use of the current models.

2. Following these principles, the authors trained a prompt generator Res-Captioner. To achieve this, they use GPT4 to generate more prompts and using human perception to filter the most productive prompt to build a dataset. This approach is practical.

3. This paper presents a new real world test set RealIR.

**Weaknesses:**

There is nothing wrong with the method of the article, which is practical and feasible. The biggest problem with the paper is its motivation, its ultimate goal is to generate better prompt for existing models, so the overall contribution is limited.

1. First, the paper does not improve the generation performance, or generalization, of existing models. The upper limit of the model's capability has been fixed by SUPIR or StableSR.

2. This work is more of a guide to the use of the current model. Although the article suggests some guidelines for the use of prompt, every researcher or user knows that better descriptions lead to better generation, and this is not a new insight. It brings little new academic insights. Further, models like SUPIR themselves allow users to adjust prompt inputs. I don't think the Res-Captioner is better or more appropriate to users' needs than a their own prompt input. (The article also does not provide the relevant exploration.)

3. The actual implementation of these guidelines is nothing new, the key is to collect the data and train MLLM, and it is model specifc. For SPUIR we need to annotate the data and train the Res-Captioner, and for StableSR we need additional data collection and finetune. So, in my opinion, this method is too engineering.

4. Again, because the goal is to use the existing model, the obervation and prompt principle are relatively shallow and easy to be changed. For example, the damage caused by quality-related prompt words comes from the fact that the training of stable diffusion may not target the low-level vision, which cen be updated. If there is a new work that solve this problem, so that the model can understand the low-level visual prompt, the observations in the article will have a new change, which is what I said, this work is like a product manual, if the product changes, the manual will fail.

5. Finally, the uesd metrics are both semantic and perceptive, so it's not surprising that when more specific prompt is provided, the generated model necessarily yields better metrics. Of course, this problem should also be attributed to the current lack of good evaluation indicators. Compared with the previous motivation problem, it is acceptable.

**Questions:**

Please see weaknesses.

---

### Official Review · Reviewer_XuM6 · 2024-11-03

**Soundness:** 2
**Presentation:** 2
**Contribution:** 2
**Rating:** 3
**Confidence:** 4

**Summary:**

The paper notes that off-the shelf captioning models do not generate detailed captions of optimal length for degraded images, and may often generate harmful captions which affect restoration tasks. To tackle this, this paper introduces Res-Captioner, a plug-and-play module to generate detailed captions tailored to both image content and degradation type for text-guided image restoration.
This captioner is obtained by fine-tuning LLaVA-1.5 model using low-rank adaptation and using chain of thought captioning to predict optimal number of tokens.  The captions generated by the proposed Res-Captioner  improve the visual quality of results produced by text guided restoration models.

**Strengths:**

The idea of improving captions to improve text guided restoration is sensible.
The restoration results look realistic and visually pleasing in comparison with the baselines.

**Weaknesses:**

The proposed approach seems very complex, combining several models to achieve the proposed results. The method requires a finetuned LLaVa model, a CLIP visual encoder, an adapter, a degradation-aware visual encoder, a tokenizer, just for caption generation.The generated caption is then used in conjunction with a text based image restoration method such as Stable SR, SUPIR etc.
Training data generation itself requires generating multiple prompts from GPT4 and human annotators selecting optimal text input providing best result, this cannot be scaled up.  Does a single caption provide the best result across different restoration modules for a given LQ input, or is the best caption diffusion model specific?

It is not clear what the authors refer to as 'ood'. Do they mean the low quality images are out of distribution for the captioning model? Or the restoration pipeine is trained for specific degradations, and tested on a more varied set of degradations? l189. mentions the baselines SUPIR and StableSR  as being trained on 4× Real-ESRGAN setting.
l337 mentions that restoration module StableSR is again finetuned with new data.This makes comparison unfair, as the improvements seen are not due to the Res-Captioner alone (which was already trained with input text which provides perceptually best quality results)

The  paper introduces complicated terminology which are not well-defined, for instance, generative capability inactivation,  generative capability deactivation, auxiliary invariant representation. The terminology makes the text a hard read.



There are some issues with novelty and The paper misses discussion on some recent relevant literature:
SUPIR already uses MLLM generated captions and uses degradation robust visual encoder in its pipeline.
SeeSR also propose to use degradation aware prompts in text based image restoration.
[1] introduced degradation-aware CLIP for image restoration, which classifies degradation types and adapts the fixed CLIP image encoder to output high-quality embeddings from degraded image inputs for image restoration. This work is very relevant to the current submission which also uses degradation aware visual encoder. Further, the idea of degradation aware prompts and degradation aware embeddings have some similarity.
[2] outperforms both SUPIR and SeeSR both on pixel wise distortion and perceptual metrics, and was demonstrated on different restoration tasks such as realworld SR, denoising, lowlight enhancement, rain drop removal.
Though not in the context of restoration, there are previous works such as [3] that considered optimizing prompts for text to image generation.
[5]design a visual-to-text adapter to adapt the embedding of low-quality images from the image domain to the textual domain as the textual guidance for SD, for improved texture generation in restoration tasks.

Sections 2.1.2 and 2.1.3 are quite obvious. Replacing relevant text by irrelevant words surely affects text-to-image generation/restoration. Similarly, it is obvious that descriptions like blur, softer backdrop, bokeh, prevent recovery of sharp imges. Previous works on text guided image restoration such as [4] as well as SUPIR use negative prompts in classifier free guidance to avoid such effects, SUPIR specifically includes these negative samples during training to ensure the model learns negative-quality concepts.

It is not clear if the RealIR benchmark would be made public.

References:

[1] Luo etal.  Controlling vision-language models for universal image restoration. In ICLR 2024.
[2] Luo etal. Photo-Realistic Image Restoration in the Wild with Controlled Vision-Language Models. In CVPR Workshops 2024.
[3] Hao et al. Optimizing Prompts for Text-to-Image Generation, In NeurIPS 2023
[4] Chandramouli etal. Text-guided Explorable Image Super-resolution, In CVPR 2024
[5] Ren etal. MoE-DiffIR: Task-Customized Diffusion Priors for Universal Compressed Image  Restoration. In ECCV 2024

**Questions:**

Does a single caption provide the best result across different restoration modules for a given LQ input, or is the best caption diffusion model specific?


T2I models are known to produce widely different outputs for different random initialization. What is the effect of different random seeds on the proposed pipeline?



As the paper considers ood generalization, it is a good idea to show generalization to diverse degradations. For instance SUPIR paper considers 4x SR 8x SR, mixture of degradations including blur, low resolution, noise and jpeg compression. Generalization to different degradations without retraining Res-Captioner would have made a stronger case.


Also see other points mentioned in the weaknesses section

---

### Note · Authors · 2024-11-14

I have read and agree with the venue's withdrawal policy on behalf of myself and my co-authors.